# Analysis of Stray Light and Enhancement of SNR in DMD-Based Spectrometers

**DOI:** 10.3390/s22166237

**Published:** 2022-08-19

**Authors:** Xiangzi Chen, Xiangqian Quan

**Affiliations:** 1College of Marine Science and Technology, Hainan Tropical Ocean University, Sanya 572022, China; 2Institute of Deep-Sea Science and Engineering, Chinese Academy of Sciences, Sanya 572000, China

**Keywords:** stray light analysis, digital micro-mirror device (DMD)-based spectrometer, data processing, Hadamard transform (HT)

## Abstract

Due to advantages such as the high efficiency of light utilization, small volume, and vibration resistance, digital micro-mirror device (DMD)-based spectrometers are widely used in ocean investigations, mountain surveys, and other field science research. In order to eliminate the stray light caused by DMDs, the stray light in DMD-based spectrometers was first measured and analyzed. Then, the stray light was classified into wavelength-related components and wavelength-unrelated components. Moreover, the noise caused by the stray light was analyzed from the perspective of encoding equation, and the de-noising decoding equation was deduced. The results showed that the accuracy range of absorbance was enhanced from [0, 1.9] to [0, 3.1] in single-stripe mode and the accuracy range of absorbance was enhanced from [0, 3.8] to [0, 6.3] in Hadamard transform (HT) multiple-stripe mode. A conclusion can be drawn that the de-noising strategy is feasible and effective for enhancing the SNR in DMD-based spectrometers.

## 1. Introduction

### 1.1. The Background of the Enhancement of SNR in DMD-Based Spectrometers

As novel digital transform spectrometers, the digital micro-mirror device (DMD)-based Hadamard transform (HT) spectrometers have been widely used in ocean investigation and other field science studies because of advantages such as the high efficiency of light utilization, the resistance to vibration, and the small volume [1,2,3,4,5]. However, stray light caused by the micro-mirrors lowered the signal-to-noise ratio (SNR) of the spectrometer [6,7]. Consequently, strategies for eliminating the stray light in DMD-based spectrometers have attracted more and more attention [8,9]. Carrying out a series of measurement with stray light, Kenneth D. et al. quantitatively analyzed the impact of stray light on SNR in DMD-based spectrometers. This study laid the foundation for eliminating stray light and enhancing the SNR in DMD-based spectrometers [10,11,12]. Wang X. et al. measured the diffraction of light in optical systems and enhanced the SNR in DMD-based spectrometers by optimizing the optical system of spectrometer [13,14,15]. Zhang Zhihai et al. enhanced the SNR in DMD-based spectrometers by changing the Hadamard transform order [16,17]. Rasmussen et al. introduced an absorption pool to reduce the stray light caused by the “off” state micro-mirrors, but the stray light and background light beyond the absorption pool still existed [18,19]. Quan X. et al. presented a new system of compound parabolic concentrator to suppress the stray light of micro-mirrors beyond the acceptance angle. However, stray light still existed within the acceptance angle [20,21].

### 1.2. Our Work

In this paper, based on the measurement and analysis of the stray light, the stray light was classified into the variable stray light related to the wavelength and the intrinsic stray light unrelated to the wavelength. Moreover, the impact of stray light on the encoding equation was analyzed and the decoding equation of eliminating stray light was deduced. Finally, the absorbance was corrected both in single-stripe mode and HT multiple-stripe mode. The results showed that the accurate range of absorbance was enhanced from [0, 1.9] to [0, 3.1] in the single-stripe mode, and the accurate range of corrected absorbance was enhanced from [0, 3.8] to [0, 6.3] in the HT multiple-stripe mode.

## 2. The Analysis and Classification of Stray Light in DMD-Based Spectrometers

The stray light in DMD-based spectrometers is mainly divided into the diffraction of DMD, the reflection light of micro-mirrors on “off” state, the reflection light caused by mechanical structure, the background light, and so on. The prototype of DMD-based spectrometer is shown in Figure 1. The polychromatic light from fiber is collimated by the collimating lens. Collimated light is split by the grating and focused onto the DMD by the imaging lens. After encoding with the DMD according to the modulation mode, the different spectrum components are concentrated onto the single detector by the converging lens. Eventually, the spectra are decoded by the computer. DMD consist of 1024 × 768 micro-mirrors with a pixel size of 13.68 μm × 13.68 μm and tilt angle of ±12° mounted on a 14.68 μm × 14.68 μm pitch. The dimension of grating is 12.8 mm × 6.4 mm with a groove density of 300 lines/mm. The light source is a near-infrared lamp with the spectrum range covering from 1.35 to 2.45 μm; then, the light is passed through the fiber with a numerical aperture of 0.2. The light source power is 12 W. The detector is an InGaAs detector with area of 2 mm^2^. The size of the spectrometer is 150 × 150 × 120 mm^3^. The stray light $I_{\mathrm{off}}$ (when all the micro-mirrors are in the “off” state) and the signal light $I_{\mathrm{on}}$ (when all the micro-mirrors are in the “on” state) were measured by inserting the filters in the sampling pool. Moreover, by changing the light source intensity, the relationships between stray light $I_{\mathrm{off}}$ with signal light $I_{\mathrm{on}}$ were derived.

The relationships of $I_{\mathrm{on}}$ and $I_{\mathrm{off}}$ at wavelengths of 1.44 μm, 1.72 μm, 1.92 μm, 2.10 μm, and 2.29 μm were measured, as shown in Figure 2.

As shown in Figure 2, with the change in light source intensity, the relationship between $I_{\mathrm{off}\text{\_}i}$ and $I_{\mathrm{on}\text{\_}i}$ is linear. Here, *i* represents the serial number at different wavelengths. The relationship between $I_{\mathrm{off}\text{\_}i}$ and $I_{\mathrm{on}\text{\_}i}$ is fitted with a linear function, as shown in Equation (1).
(1)$$I_{\mathrm{off}\text{\_}i}=\delta_{i}\cdot I_{\mathrm{on}\text{\_}i}+\varepsilon_{i}$$
where $\delta_{i}$ is the gradient of the fitted linear function which corresponds to the variable stray light related to the wavelength. $\varepsilon_{i}$ is the intercept of the fitted linear function at different wavelengths, which corresponds to the intrinsic stray light unrelated to the wavelength. The fitted linear function is written in the form of a matrix as Equation (2):(2)$$I_{\mathrm{off}}=\delta \cdot I_{\mathrm{on}}+\varepsilon$$
where $I_{\mathrm{off}}={\left[I_{\mathrm{off}\text{\_}1}\;I_{\mathrm{off}\text{\_}2}\;\cdots \;I_{\mathrm{off}\text{\_}n}\right]}^{T}$ is the stray light matrix composed of $I_{\mathrm{off}\text{\_}i}$, $I_{\mathrm{on}}={\left[I_{\mathrm{on}\text{\_}1}\;I_{\mathrm{on}\text{\_}2}\;\cdots \;I_{\mathrm{on}\text{\_}n}\right]}^{T}$ is the signal light matrix composed of $I_{\mathrm{on}\text{\_}i}$, $\delta ={\left[\delta_{1}\;\delta_{2}\;\cdots \;\delta_{n}\right]}^{T}$ is the matrix composed of the ratio coefficients of variable stray light to signal light, and $\varepsilon ={\left[\varepsilon_{1}\;\varepsilon_{2}\;\cdots \;\varepsilon_{n}\right]}^{T}$ is the matrix composed of intrinsic stray light.

According to the ratio of variable stray light to signal light at the sampling wavelength, the fitting curve of the variable stray light ratio is shown in Figure 3.

Corresponding to the code equation, the ratio of the variable stray light to signal light at the entire spectrum band is extended as Equation (3) in the matrix form:(3)$$\delta^{\prime }=\left(\begin{array}{c} \delta_{1}\;\delta_{2}\;\cdots \;\delta_{N}\\ \delta_{1}\;\delta_{2}\;\cdots \;\delta_{N}\\ \;\vdots \;\vdots \;\ddots \;\vdots \;\\ \delta_{1}\;\delta_{2}\;\cdots \;\delta_{N} \end{array}\right)$$
where *N* represents the time of measurement which corresponds to the sampling wavelength. The intrinsic stray light matrix is shown in Equation (4):(4)$$\varepsilon^{\prime }=\left(\begin{array}{c} \bar{\varepsilon }\\ \bar{\varepsilon }\\ \;\vdots \;\\ \bar{\varepsilon } \end{array}\right)$$
where $\bar{\varepsilon }$ is the average value of the measured intrinsic stray light.

## 3. The Impact of Stray Light on the Encoding Equation and Decoding Equation

There are two common coding modes: single-stripe mode and Hadamard multiple-stripe mode; the single-stripe mode has the advantages of simplicity and the Hadamard multiple-stripe mode has advantage of a high SNR. HT spectrometers boost the SNR according to specific encoding patterns. The H-matrix, the S-matrix, and the complementary S-matrix boost the SNR by factors of $\sqrt{N}$, $\frac{N+1}{2\sqrt{N}}$, and $\frac{\sqrt{N+1}}{2}$, respectively, where *N* is the Hadamard order. In this paper, we adopted the S-matrix in the Hadamard transform coding mode.

### 3.1. The Impact of Stray Light in Single-Stripe Mode

There are two types of spectrum acquisition modes in DMD-based spectrometers: single-stripe mode and multiple-stripe mode. The single-stripe coding process is shown in Figure 4.

Ideally, the single-stripe scanning process is expressed in a matrix, such as in Equation (5):(5)$$\left(\begin{array}{c} I_{1}\\ I_{2}\\ \vdots \\ I_{N} \end{array}\right)=\left(\begin{array}{c} 1\;0\;\cdots \;0\\ 0\;1\;\cdots \;0\\ \vdots \;\vdots \;\ddots \;\vdots \\ 0\;0\;\cdots \;1 \end{array}\right)\left(\begin{array}{c} E_{1}\\ E_{2}\\ \vdots \\ E_{N} \end{array}\right)$$
where $E_{1}$–$E_{N}$ is the spectral intensity unfolded on the DMD. $I_{1}$–$I_{N}$ is the detected light intensity. The encoding matrix is a unit matrix. The number of matrix columns represents the time of the measurements, and the number of matrix rows represents the value of the measured wavelength. The element “1” in the matrix indicates that the micro-mirror is in the “on” state. The element “0” in the matrix indicates that the micro-mirror is in the “off” state. The encoding matrix could be described as Equation (6) in a simplified form:(6)I = U×E
where I is the detected light intensity matrix, E is the spectral intensity matrix, and U is the N-order unit matrix for coding. However, due to the existence of stray light, the element “0” in the encoding matrix may be greater than 0. Based on the variable stray light ratio matrix (3), the spectral encoding matrix with variable stray light is shown as Equation (7).
(7)$$\left(\begin{array}{c} I_{1}\\ I_{2}\\ \vdots \\ I_{N} \end{array}\right)=\left(\begin{array}{c} 1\;0\;\cdots \;0\\ 0\;1\;\cdots \;0\\ \;\vdots \;\vdots \;\ddots \;\vdots \\ 0\;0\;\cdots \;1 \end{array}\right)\left(\begin{array}{c} E_{1}\\ E_{2}\\ \vdots \\ E_{N} \end{array}\right)+\left(\left(\left(\begin{array}{c} 1\;1\;\cdots \;1\\ 1\;1\;\cdots \;1\\ \;\vdots \;\vdots \;\ddots \;\vdots \\ 1\;1\;\cdots \;1 \end{array}\right)-\left(\begin{array}{c} 1\;0\;\cdots \;0\\ 0\;1\;\cdots \;0\\ \;\vdots \;\vdots \;\ddots \;\vdots \\ 0\;0\;\cdots \;1 \end{array}\right)\right)\cdot \left(\begin{array}{c} \delta_{1}\;\delta_{2}\;\cdots \;\delta_{N}\\ \delta_{1}\;\delta_{2}\;\cdots \;\delta_{N}\\ \;\vdots \;\vdots \;\ddots \;\vdots \;\\ \delta_{1}\;\delta_{2}\;\cdots \;\delta_{N} \end{array}\right)\right)\left(\begin{array}{c} E_{1}\\ E_{2}\\ \vdots \\ E_{N} \end{array}\right)$$

The equation could be expressed in a simplified form as Equation (8).
(8)$$I_{\delta }=\;U\times E+\left(O-U\right)\cdot \delta^{\prime }\times E$$
where $I_{\delta }$ is the detected light intensity matrix with variable stray light, and O is the N-order square matrix with all elements as “1”. Based on the intrinsic stray light in DMD-based spectrometers, the encoding equation is modified to Equation (9).
(9)$$\left(\begin{array}{c} I_{1}\\ I_{2}\\ \vdots \\ I_{N} \end{array}\right)=\left(\begin{array}{c} 1\;0\;\cdots \;0\\ 0\;1\;\cdots \;0\\ \;\vdots \;\vdots \;\ddots \;\vdots \\ 0\;0\;\cdots \;1 \end{array}\right)\left(\begin{array}{c} E_{1}\\ E_{2}\\ \vdots \\ E_{N} \end{array}\right)+\left(\left(\left(\begin{array}{c} 1\;1\;\cdots \;1\\ 1\;1\;\cdots \;1\\ \;\vdots \;\vdots \;\ddots \;\vdots \\ 1\;1\;\cdots \;1 \end{array}\right)-\left(\begin{array}{c} 1\;0\;\cdots \;0\\ 0\;1\;\cdots \;0\\ \;\vdots \;\vdots \;\ddots \;\vdots \\ 0\;0\;\cdots \;1 \end{array}\right)\right)\cdot \left(\begin{array}{c} \delta_{1}\;\delta_{2}\;\cdots \;\delta_{N}\\ \delta_{1}\;\delta_{2}\;\cdots \;\delta_{N}\\ \;\vdots \;\vdots \;\ddots \;\vdots \;\\ \delta_{1}\;\delta_{2}\;\cdots \;\delta_{N} \end{array}\right)\right)\left(\begin{array}{c} E_{1}\\ E_{2}\\ \vdots \\ E_{N} \end{array}\right)+\left(\begin{array}{c} \bar{\varepsilon }\\ \bar{\varepsilon }\\ \;\vdots \;\\ \bar{\varepsilon } \end{array}\right)$$

The equation could be expressed in a simplified form as Equation (10):(10)$$I_{\delta +\varepsilon }=\;U\times E+\left(O-U\right)\cdot \delta^{\prime }\times E+\varepsilon^{\prime }$$
where ${Ι}_{\delta +\varepsilon }$ is the detected light intensity matrix with two types of stray light. The decoded spectral intensity matrix could be derived as Equation (11):(11)$$E_{\delta +\varepsilon }=U^{-1}\times I_{\delta +\varepsilon }\;=E+U^{-1}\times \left(\left(O-U\right)\cdot \delta^{\prime }\times E+\varepsilon^{\prime }\right)=E+\left(O-U\right)\cdot \delta^{\prime }\times E+\varepsilon^{\prime }$$
where $E_{\delta +\varepsilon }$ is the spectral intensity matrix with two types of stray light. The spectral intensity matrix eliminating two types of stray light can be expressed as Equation (12):(12)$$E=E_{\delta +\varepsilon }-\left(O-U\right)\cdot \delta^{\prime }\times E+\varepsilon^{\prime }$$

To simplify the calculation, we substituted $E\approx E_{\delta +\varepsilon }$ into Equation (12). Then, the decoding matrix could be derived as Equation (13):(13)$$E\approx E_{\delta +\varepsilon }-\left(O-U\right)\cdot \delta^{\prime }\times E_{\delta +\varepsilon }+\varepsilon^{\prime }$$

### 3.2. The Impact of Stray Light in Hadamard Multiple-Stripe Mode

The Hadamard coding process of DMD-based spectrometers is shown in Figure 5.

Corresponding to the single-stripe mode, the encoding matrix is changed from U to H. Accordingly, the encoding matrix with two types of stray light is shown as Equation (14):(14)$$I_{\delta +\varepsilon }=\;H\times E+\left(O-H\right)\cdot \delta^{\prime }\times E+\varepsilon^{\prime }$$
where H is the Hadamard encoding matrix. The decoding matrix eliminating two types of stray light is shown as Equation (15):(15)$$E_{\delta +\varepsilon }=H^{-1}\times I_{\delta +\varepsilon }\;=\;E+H^{-1}\times \left(\left(O-H\right)\cdot \delta^{\prime }\times E\right)+H^{-1}\times \varepsilon^{\prime }$$

Then, the decoding matrix eliminating two types of stray light can be shown as Equation (16):(16)$$E=\;E_{\delta +\varepsilon }-H^{-1}\times \left(\left(O-H\right)\cdot \delta^{\prime }\times E\right)+H^{-1}\times \varepsilon^{\prime }$$

To simplify the calculation, we substituted $E\approx E_{\delta +\varepsilon }$ into (16); then, the decoding matrix was derived as Equation (17):(17)$$E\approx E_{\delta +\varepsilon }-H^{-1}\times \left(\left(O-H\right)\cdot \delta^{\prime }\times E_{\delta +\varepsilon }\right)+H^{-1}\times \varepsilon^{\prime }$$

## 4. Experiments and Methods

To certify the efficiency of our SNR enhancement strategy in DMD-based spectrometers, we carried out a series of experiments and simulations. The process and method are shown as follows:

Firstly, we assumed that the light source spectrum was the constant of “1”, whereas the ideal absorption spectrum was a normal curve. With this assumption, we could obtain the referential spectrum and absorbance.Secondly, the Hadamard transform order was set to 255 during the data processing. Based on the stray light detected in the experiment (as shown in Figure 2), the spectrum with the noise of stray light was calculated and the absorbance with the noise could be derived in both single-stripe mode and Hadamard multiple-stripe mode. Consequently, we could obtain the raw spectrum and absorbance data with the noise. By contrasting the raw spectrum and absorbance with the referential spectrum and absorbance, the effect of the noise could be calculated quantitatively. We defined that the accuracy of absorbance was less than 0.1, then the accuracy range could be derived.Thirdly, based on the decoding matrix of eliminating the two types of stray light, the spectrum and absorbance of eliminating the stray light noise were derived. By contrasting the corrected spectrum and absorbance with the referential spectrum and absorbance, the accuracy range could be derived both in the single-stripe mode and the Hadamard multiple-stripe mode, respectively.Finally, by contrasting the accuracy range of absorbance between raw absorbance with corrected absorbance, the efficiency of this strategy was certified.

## 5. The Result of Eliminating the Stray Light

### 5.1. Single-Stripe Mode

Figure 6 shows the impact of stray light on the spectrum in the single-stripe mode. The light source spectrum was assumed to have a constant value of “1”, while the ideal absorption spectrum was a normal curve (a in Figure 6). Then, the spectrum with wavelength-related stray light is shown as curve b (in Figure 6) and the spectrum with two kinds of stray light is shown as curve c (in Figure 6).

The calculation of the absorbance is shown as Equation (18):(18)AU=lg(1/T)
where T is the transmittance of the spectrum. The absorbance under different circumstances is shown in Figure 7. The ideal absorbance is shown as a (in Figure 7). The absorbance with wavelength-related stray light is shown as b (in Figure 7). The absorbance with two kinds of stray light is shown as c (in Figure 7). Assuming that the deviation between the calculated absorbance and the standard absorbance was defined to be less than 0.1, it could be derived that the accurate range was [0, 1.9].

Based on Equation (13), the corrected spectrum is derived as shown in Figure 8. Additionally, the corrected absorbance is shown as c (in Figure 8). The ideal absorbance is shown as a (in Figure 8). The absorbance with two kinds of stray light is shown as b (in Figure 8). The results showed that the accuracy range was enhanced to [0, 3.1].

### 5.2. The Hadamard Multiple-Stripe Mode

Figure 9 shows the impact of stray light on the spectrum in the Hadamard mode. The ideal absorption spectrum is a normal curve as a (in Figure 9). The spectrum with wavelength-related stray light is shown as curve b (in Figure 9). The spectrum with two kinds of stray light is shown as curve c (in Figure 9).

The absorbance values in different scenarios are shown in Figure 10. The ideal absorbance is shown as a. The absorbance with wavelength-related stray light is shown as b. The absorbance with two kinds of stray light is shown as c. It could be derived that the accurate range was [0, 3.8].

The correction of absorbance in Hadamard mode is shown in Figure 11. Curve a is the standard absorbance. Curve b is the absorbance with stray light. Compared with the single-stripe acquisition mode, the range of absorbance accuracy in Hadamard acquisition mode is [0, 3.8]. Based on the decoding matrix in Equation (17), the corrected absorbance could be calculated as curve c. The range of absorbance accuracy was enhanced from [0, 3.8] to [0, 6.3].

## 6. Conclusions

The aim of this study was to eliminate the stray light in DMD-based spectrometers. Based on experiments measuring stray light, it was classified into two types, including variable stray light related to the wavelength, and the intrinsic stray light unrelated to the wavelength. Then, the impacts of stray light on spectrum were analyzed from the perspective of the encoding equation, and the decoding equation of eliminating stray light was derived. Finally, the spectrum absorbance accurate range was enhanced from [0, 1.9] to [0, 3.1] in the single-stripe mode, and the spectrum absorbance accuracy range was enhanced from [0, 3.8] to [0, 6.3] in the Hadamard multiple-stripe mode. A conclusion can be drawn that the denoising strategy is feasible and effective for enhancing the SNR in DMD-based spectrometers.

## Figures and Tables

**Figure 1 sensors-22-06237-f001:**
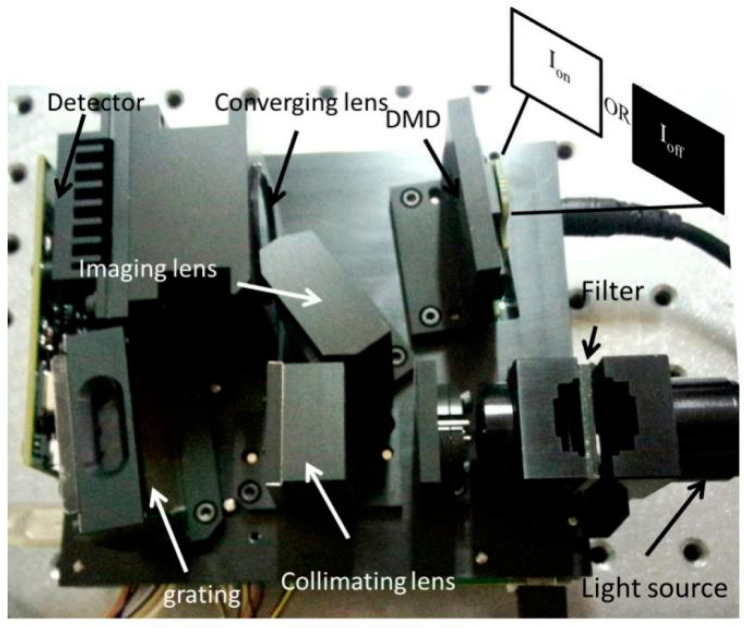
The prototype of stray light measurement in a DMD-based spectrometer: the light from the light source passes through the collimating lens, grating, imaging lens, DMD, and converging lens to the detector. When all the micro-mirrors are in the “on” state and “off” state, the detected light signals $I_{\mathrm{on}}$ and $I_{\mathrm{off}}$ can be derived, respectively.

**Figure 2 sensors-22-06237-f002:**
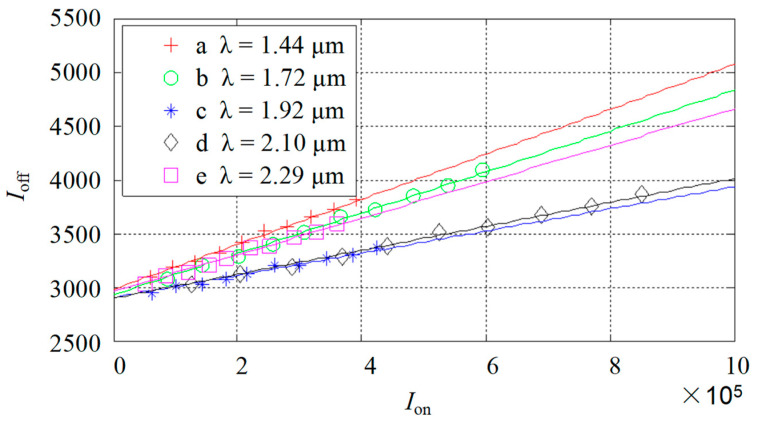
The measurement results of stray light: $I_{\mathrm{on}}$ and $I_{\mathrm{off}}$ represent the detected light signals when all the micro-mirrors are in the “on” state and “off” state, respectively. Curves a, b, c, d, and e represent the relationships of $I_{\mathrm{on}}$ and $I_{\mathrm{off}}$ when the wavebands of the inserted filter are 1.44 μm, 1.72 μm, 1.92 μm, 2.10 μm, and 2.29 μm, respectively.

**Figure 3 sensors-22-06237-f003:**
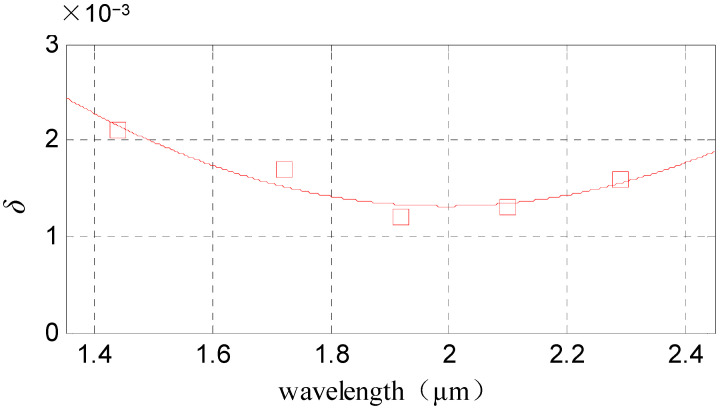
The fitting curve of variable stray light ratio: the horizontal ordinate shows the wavelength with units in micrometers. Additionally, the longitudinal coordinate shows the ratio of variable stray light to signal light, which is represented by “*δ*” in this paper.

**Figure 4 sensors-22-06237-f004:**
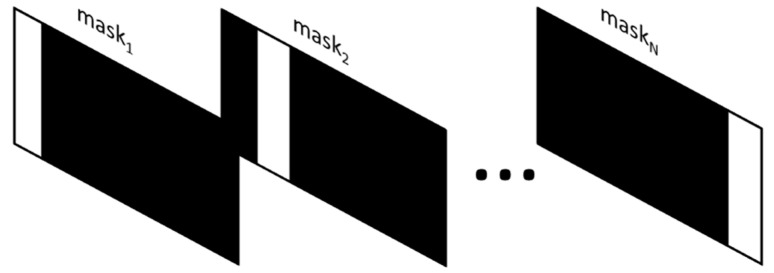
The sketch map of the single-stripe mode: in the single-stripe mode, only one column of micro-mirrors is turned on, corresponding to the single encoding matrix. The first detected light signal is reflected by mask_1_. Accordingly, the second detected light signal is reflected by mask_2_, and so on. After the last detected signal is detected with the reflection of mask_N_, the spectra can be derived.

**Figure 5 sensors-22-06237-f005:**
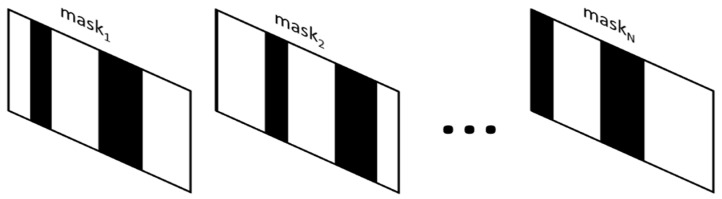
The sketch map of Hadamard multiple-stripe mode: in Hadamard multiple-stripe mode, multiple columns of micro-mirrors are turned on, corresponding to the Hadamard encoding matrix. The first detected light signal is reflected by mask_1_. Accordingly, the second detected light signal is reflected by mask_2_, and so on. After the last detected signal is detected with the reflecting of mask_N_, the raw data of the spectrum are acquired. By decoding with the computer, the spectra can be derived.

**Figure 6 sensors-22-06237-f006:**
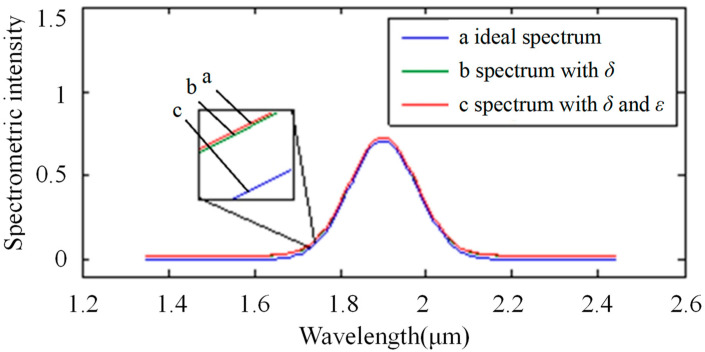
Impact of stray light on the spectrum in single-stripe mode: the horizontal ordinate shows the wavelength with units of micrometers. Additionally, the longitudinal coordinate shows the spectrometric intensity. Curve a shows the ideal spectrum, curve b shows the spectrum with the noise of variable stray light, and curve c shows the spectrum with the noise of two types of stray light, including variable stray light and intrinsic stray light.

**Figure 7 sensors-22-06237-f007:**
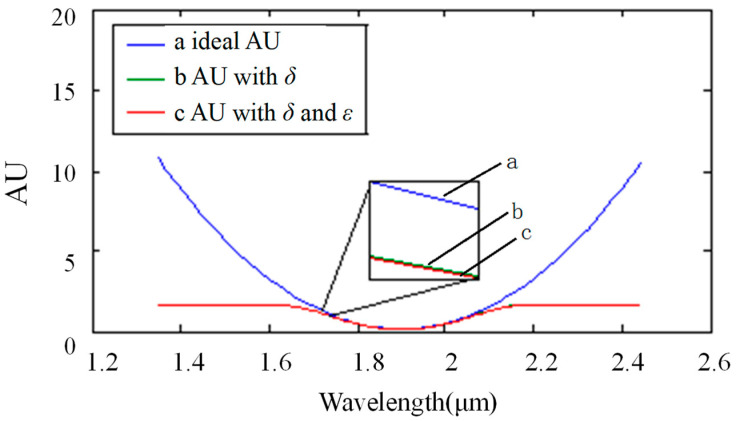
Result of stray light on the absorption spectrum in single-stripe mode: the horizontal ordinate shows the wavelength with units of micrometers. The longitudinal coordinate shows absorbance, which is represented by “AU” in this paper. Curve a shows the ideal absorbance, curve b shows the absorbance with the noise of variable stray light, and curve c shows the absorbance with the noise of two types of stray light, including variable stray light and intrinsic stray light.

**Figure 8 sensors-22-06237-f008:**
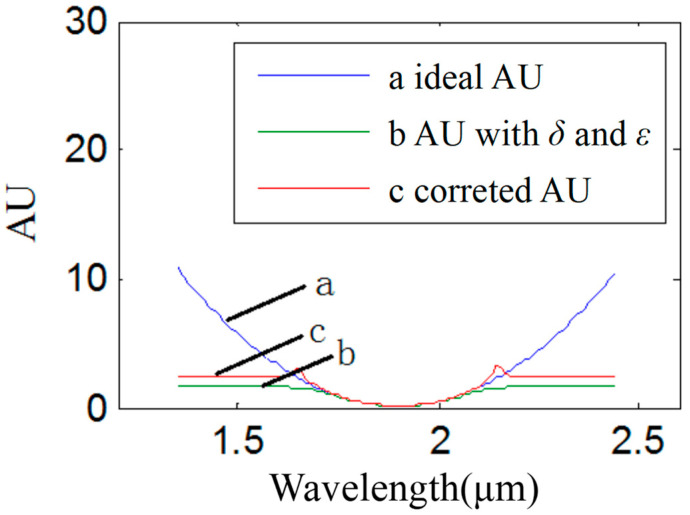
Results of absorbance correction in single-stripe mode: the horizontal ordinate shows the wavelength with units of micrometers. The longitudinal coordinate shows absorbance, which was represented by “AU” in this paper. Curve a shows the ideal absorbance, curve b shows the absorbance with the noise of two types of stray light, including variable stray light and intrinsic stray light, and curve c shows the corrected absorbance.

**Figure 9 sensors-22-06237-f009:**
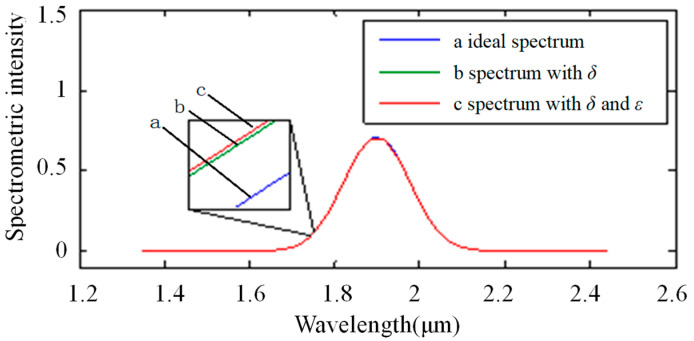
Result of stray light on the absorption spectrum in Hadamard multiple-stripe mode: the horizontal ordinate shows the wavelength with units of micrometers. Additionally, the longitudinal coordinate shows the spectrometric intensity. Curve a shows the ideal spectrum, curve b shows the spectrum with the noise of variable stray light, and curve c shows the spectrum with the noise of two types of stray light, including variable stray light and intrinsic stray light.

**Figure 10 sensors-22-06237-f010:**
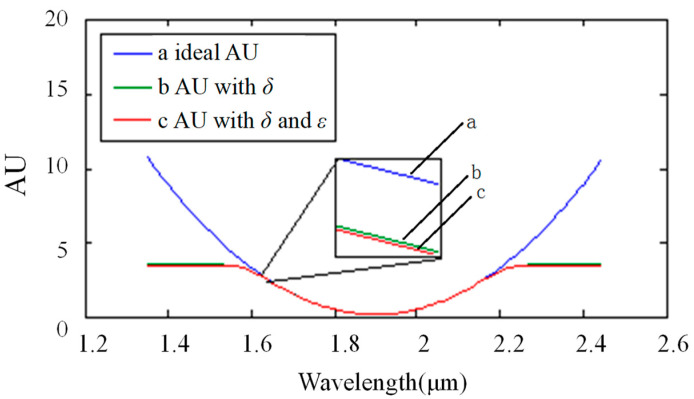
Result of stray light on the absorbance in Hadamard multiple-stripe mode: the horizontal ordinate shows the wavelength with units of micrometers. The longitudinal coordinate shows absorbance, which was represented by “AU” in this paper. Curve a shows the ideal absorbance, curve b shows the absorbance with the noise of variable stray light, and curve c shows the absorbance with the noise of two types of stray light, including variable stray light and intrinsic stray light.

**Figure 11 sensors-22-06237-f011:**
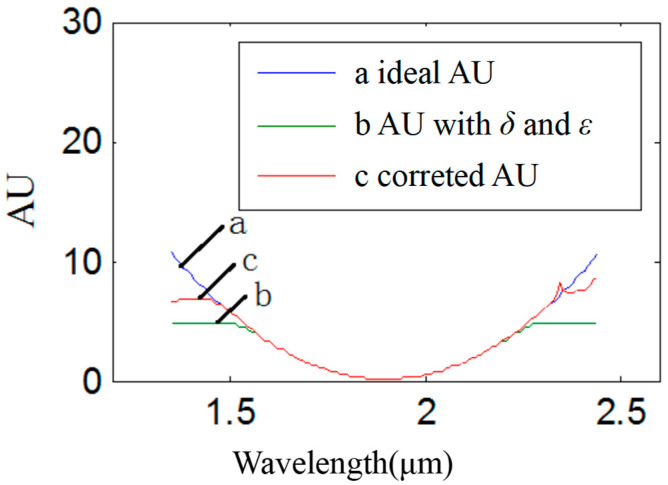
Results of absorbance correction in Hadamard mode: the horizontal ordinate shows the wavelength with units of micrometers. The longitudinal coordinate shows the absorbance, which was represented by “AU” in this paper. Curve a shows the ideal absorbance, curve b shows the absorbance with the noise of two types of stray light, including variable stray light and intrinsic stray light, and curve c shows the corrected absorbance.

## Data Availability

The data presented in this study are available from the corresponding author upon request.

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
