# Peer review of "Analysis of Stray Light and Enhancement of SNR in DMD-Based Spectrometers"

_sensors, 2022, doi:10.3390/s22166237_

Round 1

Reviewer 1 Report

This paper presented an interesting strategy to enhance the accurate range of absorbance in DMD based spectrometer. Firstly, the stray light was classified into the variable stray light and the intrinsic stray light. Then, the impact of stray light on the encoding equation was analyzed and the decoding equation of eliminating stray light was deduced. The method was certified to be effective both in the single stripe mode and HT multiple stripes mode. As far as I am concerned, This is a well-written paper containing interesting results which merit publication. For the benefit of the readers, some minor problems should be improved as follows: 

1. The coding process in figure 4 and figure 5 should be describe in detail.

2. Some writing problems should be improved such as the impact in Figure 6 should be corrected to Impact, and the word Where under the equation 1 should corrected to where and no indent required.

3. As you mentioned, there were two coding modes. Why you chose these two modes? You should describe the Hadamard coding mode in brief.

4. You should add some parameters about your experimental device for the readers. Just like the information about the DMD, the light source and detector, etc

Reviewer 2 Report

The reviewed work is very interesting. Based on the measurement and analysis of  the stray light, the authors classified of the stray light as variable scattered light related to the wavelength, and intrinsic scattered light is not related to the wavelength. The authors corrected the absorbance in both single strip mode and HT multi-strip mode. The exact range of absorbance was increased from [0.1.9] to [0.3.1] in single strip mode and the exact range of corrected absorbance was increased from [0.3.8] to [0.6.3] in multi-HT strip mode. Overall, I assess the manuscript positively. However, the manuscript needs to be revised. 

1. Authors must add the 'Experimetal and Methods' chapter and describe in detail the methodology of the performed research.

2. The conclusions consistent with the evidence and arguments presented  are address the main question posed in manuscript.

3. The cited literature is appropriate.

4. Under the description of each figures, there must be a legend explaining the abbreviations and possible names. 
